# Predictors of Outcome in Patients with Pulmonary Hypertension Undergoing Mitral and Tricuspid Valve Surgery

**DOI:** 10.3390/medicina59061103

**Published:** 2023-06-07

**Authors:** Ee Phui Kew, Vincenzo Caruso, Julia Grapsa, Paolo Bosco, Gianluca Lucchese

**Affiliations:** 1Department of Cardiac Surgery, St. Thomas Hospital London, London SE1 7EH, UKgianluca.lucchese@gmail.com (G.L.); 2Department of Cardiology, St. Thomas Hospital London, London SE1 7EH, UK

**Keywords:** cardiac surgery, echocardiography, pulmonary hypertension, mitral valve, tricuspid valve

## Abstract

*Background and Objectives*: Pulmonary hypertension (PH) secondary to left-sided valvular heart disease is associated with poor cardiac surgical outcome compared with patients without PH. Our objective was to investigate the prognostic factors of surgical outcome in patients with PH undergoing mitral valve (MV) and tricuspid valve (TV) surgery, in order to risk stratify their management. *Materials and Methods*: This is a retrospective observational study on patients with PH who underwent MV and TV surgery from 2011 to 2019. The primary outcome was all-cause mortality. The secondary outcomes were post-op respiratory and renal complications, length of intensive care unit stay and length of hospital stay. *Results*: Seventy-six patients were included in this study. The all-cause mortality was 13% (*n* = 10), with mean survival of 92.6 months. Among the patients, 9.2% (*n* = 7) had post-op renal failure requiring renal replacement therapy and 6.6% (*n* = 5) had post-op respiratory failure requiring intubation. Univariate analysis demonstrated that pre-operative left ventricular ejection fraction (LVEF), peak systolic tissue velocity at the tricuspid annulus (S’) and etiology of MV disease were associated with respiratory and renal failure. Tricuspid annular plane systolic excursion (TAPSE) was associated with respiratory failure only. S’, type of operation, LVEF, urgency of surgery, and etiology of MV disease were found to be predictive of mortality. After excluding redo mitral surgery, all statistically significant findings remain unchanged, with the addition of right ventricular (RV) size being associated with respiratory failure. In the subgroup analysis of routine cases (*n* = 56), patients with primary mitral regurgitation who underwent mitral valve repair had better survival outcome. *Conclusions*: Urgency of surgery, etiology of MV disease, type of operation (replacement or repair), S’ and pre-op LVEF are prognostic indicators in this small cohort of patients with PH undergoing MV and TV surgery. A larger prospective study is warranted to validate our findings.

## 1. Introduction

Pulmonary hypertension (PH) in the context of cardiac surgery is usually caused by long-standing left-sided valvular heart disease. Chronic PH can cause right ventricular (RV) overload, which leads to RV failure. This will subsequently lead to functional tricuspid valve (TV) regurgitation.

There are relatively few published data on the management of PH secondary to valvular heart disease [1]. Its management can be broadly categorized into medical management and surgical intervention. Medical management includes non-specific PH drugs such as nitrates and hydralazine, and specific PH drugs such as endothelin receptor antagonists and phosphodiesterase-5 inhibitors [2]. Drug therapies in this group of patients have been shown to be of no benefit in some randomized controlled trials [3,4]. In fact, the sildenafil for improving outcomes after valvular correction (SIOVAC) trial revealed that the use of sildenafil in post-heart valve surgery patients led to worse clinical outcomes, including death [5]. However, Eskandr et al. reported that milrinone is an effective drug in reducing pulmonary artery pressure and pulmonary capillary wedge pressure, as well as providing adequate cardiac performance, in patients with pulmonary hypertension undergoing mitral valve surgery [6].

The aim of this study is to investigate and identify the prognostic indicators of cardiac surgical outcome in patients with PH and underlying mitral valve disease. This will help us to further risk stratify this cohort of patients and support better clinical decision making in terms of timing and patient selection for surgery.

## 2. Materials and Methods

### 2.1. Study Design and Study Population

We conducted a retrospective observational study on all patients with PH undergoing mitral valve and tricuspid valve surgery in our unit between 2011 and 2019. Inclusion criteria were adults older than 18 years old who had both mitral and tricuspid valve surgery at the same time, and proven raised pulmonary pressures on echocardiogram pre-operatively. Patients with free-flow TR and any concomitant cardiac surgery such as coronary artery bypass graft and aortic valve surgery were excluded. Infective endocarditis cases were also excluded.

The primary end point was all-cause mortality. The secondary end points were respiratory failure requiring reintubation, length of intubation, renal dysfunction requiring dialysis, the length of hospital stay and the length of intensive care unit (ICU) stay.

We received approval from our local National Health Service Foundation Trust to collect and analyze the data in this study.

### 2.2. Clinical Data

Age, gender and body mass index (BMI) were collected as baseline demographics. Other variables of interest were pre-operative co-morbidities (atrial fibrillation, pre-existing renal failure, pre-existing respiratory conditions), etiology of valvular disease (primary or secondary MR, mitral stenosis (MS), primary or secondary tricuspid regurgitation (TR), tricuspid stenosis, type of operation (MV repair/replacement, TV repair/replacement) and urgency of surgery (routine, urgent, emergency). Cumulative cardiopulmonary bypass time and cumulative cross-clamp time were also recorded as intra-operative parameters.

### 2.3. Echocardiographic (ECHO) Assessment

For right ventricular assessment, we followed a multiparametric echocardiographic approach based on the latest guidelines and previous recommendation papers [7,8]. The ECHO parameters we included in this study were peak systolic tissue velocity at the tricuspid annulus (S’), tricuspid annular plane systolic excursion (TAPSE), left ventricular ejection fraction (LVEF), degree of tricuspid regurgitation (TR), right ventricular (RV) size, RV function, RV myocardial performance index (MPI), RV fractional area change (RVFAC) and right atrial volume index.

#### 2.3.1. Qualitative Assessment

RV dilatation was assessed in the parasternal long-axis, short-axis, apical four-chamber and subcostal views. RV hypertrophy was defined as a free wall thickness of >5 mm on the apical 4-chamber view. Importantly enough, RV contractility was assessed in terms of function and regional wall motion abnormalities. As a composite score of the qualitative assessment (dilatation, hypertrophy and contractility), the RV function was divided into mild, moderate or severe.

#### 2.3.2. Quantitative Parameters

The degree of TR was evaluated according to the latest guidelines and divided into mild, moderate or severe [9,10]. RV systolic pressure was measured as a derivative from tricuspid regurgitant velocity by adding right atrial pressure. It is noted that we did not recruit patients with pulmonic stenosis and therefore there was no need to employ the modified Bernoulli equation [8].

RVFAC was calculated from end-diastolic area and end-systolic area, measured from the apical four-chamber view.

S’ was determined from the average of three TDI signals from different cardiac cycles. Myocardial performance index (MPI) was measured by tissue Doppler imaging of the RV free wall and by adding isovolumic relaxation time and isovolumic contraction time and dividing the sum by RV ejection time [8].

TAPSE is the reflection of the movement of base to apex shortening of the RV in systole and was measured from the four-chamber view. When measuring TAPSE, it was important to ensure that the entire RV is included in the view, and in particular that there is no dropout in the endocardial outline along the interventricular septum and the RV free wall. Maximal TAPSE was defined as the total excursion of the tricuspid annulus from its highest position after atrial ascent to the peak descent during ventricular systole [8].

Right atrial volume index was calculated from the apical four-chamber view or from the subcostal view, and is measured at maximum atrial volume at end-systole. The single plane area–length method was used, and right atrium volume is measured using the area and the long-axis length of the atrium.

Finally, left ventricular ejection fraction (LVEF) was measured by Simpsons when endocardial definition was satisfactory and there was no dropout [11]. All the patients included had good ECHO windows for adequate assessment.

### 2.4. Statistical Analysis

Continuous variables are presented as mean and standard deviation (SD) or median and interquartile range (IQR), while categorical variables are presented as frequencies and percentage. Cumulative probability of death was estimated using the Kaplan–Meier (KM) method. Log-rank was employed to detect survival differences between groups. Univariate analysis of data was performed using χ^2^ test, Mann–Whitney test and Wilcoxon signed rank test where appropriate. For any significant variables detected, Cox regression was used to calculate hazard ratio (HR) and to detect predictors of mortality. *p* < 0.05 was considered statistically significant.

## 3. Results

A total of 76 patients with PH underwent MV and TV surgery from May 2011 to Dec 2019. The mean age of this cohort was 66.9 ± 13.3 years old, mean BMI was 26.5 ± 4.7 kg/m^2^, and mean pre-op baseline eGFR was 63.5 ± 24.2 mL/min/1.73 m^2^. The remaining demographic details are listed in Table 1.

The mean cumulative cardiopulmonary bypass time was 135.0 min, and the mean cumulative cross-clamp time was 99.6 min. Both parameters were not associated with survival (*p* = 0.558 and *p* = 0.289, respectively). Pulmonary hypertension was assessed with ECHO measurement of RV systolic pressure with the assumption that there is no RV outflow tract obstruction. Among the patients, 9.2% (*n* = 7) had mild PH (RVSP 35–45 mmHg), 39.5% (*n* = 30) had moderate PH (RVSP 46–60 mmHg), and 51.3% (*n* = 39) had severe PH (RVSP > 60 mmHg). The degree of PH in this cohort did not affect their survival (*p* = 0.374).

### 3.1. Primary Outcome and Survival Analysis (Kaplan–Meier)

The overall all-cause mortality of this cohort was 13% (*n* = 10), with in-hospital mortality of 9.5% (*n* = 7) and 30-day all-cause mortality of 7.5% (*n* = 6). The mean survival of this cohort was 92.6 months (95% CI 84.3, 101.0 months), with 1-year and 5-year survival rates of 89.5% and 87.6%, respectively. The median follow-up was 54.5 months (interquartile range 33.3–71.3 months).

### 3.2. Secondary Outcomes

The median length of hospital stay was 8 days (IQR 6–15 days). The median ICU stay was 1 day (IQR 1–3 days). The median intubation time was 10 h (IQR 6–17 h). Among the patients, 9.2% (*n* = 7) had post-op renal failure requiring renal replacement therapy while 6.6% (*n* = 5) experienced post-op respiratory failure requiring reintubation.

### 3.3. Prognostic Factors of Survival—Univariate Analysis

The parameters which were found to be associated with survival were urgency of surgery, S’, LVEF, type of operation, right ventricular size, etiology of MV disease and degree of TR. Cox regression was then applied, and four parameters (S’, type of operation, etiology of MV disease, LVEF) demonstrated statistically significant HR.

In this cohort, there are 56 routine cases, 15 urgent cases and 5 emergency cases. There is a significant difference in survival between these three groups (*p* < 0.001). Routine cases showed the most favorable outcome with mean survival of 97.4 months (95% CI 89.3, 105.4 months), followed by urgent cases with mean survival of 74.6 months (95% CI 59.9, 89.4 months). Emergency cases showed the worst outcome with mean survival of 25.4 months (95% CI 0, 52.3 months).

The cohort with S’ of <9.5 cm/s (*n* = 15) demonstrated poorer survival compared with the group with S’ of ≥9.5 cm/s (*n* = 48), with mean survival of 57.4 months and 102.3 months, respectively; HR 4.79 (*p* = 0.041) (Figure 1).

There were three types of operation: MV repair and TV repair (*n* = 41); MV replacement and TV repair (*n* = 33); MV replacement and TV replacement (*n* = 2). Overall, there is a significant difference in survival among these three groups. MV repair and TV repair showed better survival compared with MV replacement and TV repair, with mean survival of 85.6 months and 78.9 months, respectively; HR 0.192 (*p* = 0.037). There were only two patients who had both mitral valve and tricuspid valve replacement, and their survival is not statistically significant compared with the other two groups.

Etiology of MV surgery was divided into primary MR, secondary MR, MS and previous failed mitral surgery/redo. The survival among these groups was also statistically significant (*p* < 0.001). Primary MR showed the best survival outcome with mean survival of 101.2 months (95% CI 94.8, 107.7 months). When compared with primary MR, secondary MR and MS demonstrated statistically significant poorer survival (mean survival of 11.0 months; HR 22.51 (*p* = 0.001) and mean survival of 63.4 months; HR 7.82 (*p* = 0.007), respectively). Previous failed MV surgery/redo showed mean survival of 65.8 months (95% CI 50.6, 81.0 months), but it was not statistically significant when compared with the other groups. Post hoc analyses were conducted using Fisher’s LSD and Bonferroni plus Dunnett’s. They confirmed our findings that secondary MR and MS performed poorer post-operatively in terms of mortality compared with primary MR and previous failed MV surgery/redo.

LVEF was divided into four groups: normal (LVEF > 50%), mild (LVEF 40–49%), moderate (30–39%) and severe (LVEF < 30%). Overall, there was a significant difference in terms of survival among these four groups (*p* = 0.008). Cox regression only detected statistically significant HR between severe LVEF impairment and normal LVEF with HR of 12.5 (*p* = 0.021) (Figure 2).

RV size was divided into two groups: normal and dilated. Dilated RV is associated with poor survival compared with the normal RV group, with mean survival of 69.0 months and 103.2 months, respectively, using log-rank test (*p* = 0.026). However, further analysis with Cox regression did not detect any statistically significant difference with HR of 7.54 (*p* = 0.059) (Figure 3).

Severe TR was found to have a significantly higher mortality rate compared with non-severe (mild and moderate) TR (*p* = 0.024). Cox regression showed HR of 38.37, but it was not statistically significant (*p* = 0.195). It is worth mentioning that when the degree of TR was divided into mild, moderate and severe, the survival curve among these three groups was not statistically significant (*p* = 0.07).

### 3.4. Renal Failure Requiring Dialysis—Univariate Analysis

The parameters associated with post-op renal failure requiring dialysis were urgency of surgery (*p* < 0.001), etiology of MV disease (*p* < 0.001), S’ (*p* < 0.001) and LVEF (*p* = 0.01). There was no association between pre-operative renal impairment and post-op renal failure requiring dialysis (*p* = 0.117). See Table 2 for summary.

### 3.5. Respiratory Failure Requiring Intubation—Univariate Analysis

The parameters associated with post-op respiratory failure requiring re-intubation were urgency of surgery (*p* = 0.002), etiology of MV disease (*p* < 0.001), S’ (*p* = 0.013), LVEF (*p* < 0.001) and TAPSE (*p* = 0.025). There was no association between pre-existing respiratory disease and post-op respiratory failure requiring intubation (*p* = 0.077). See Table 2 for summary.

### 3.6. Excluding Redo MV Surgery

Redo mitral valve surgery is thought to carry higher operative risks compared with first-time mitral valve operation. There is a higher risk of intraoperative bleeding with redo sternotomy, and the cardiopulmonary bypass time might be longer depending on the extent of adhesions, which might impact outcomes. For those reasons, we performed a separate analysis on patients who had first-time mitral valve operation only. After excluding previous failed MV surgery/redo (*n* = 9), all statistically significant findings remain unchanged, with the additional finding of RV size being associated with respiratory failure (*p* = 0.046). Table 3 summarizes the surgical outcome predictors.

### 3.7. Subgroup Analysis of Only Elective Cases

Subgroup analysis was performed on only elective cases (*n* = 56). All-cause mortality was 8.9% (*n* = 5), with in-hospital mortality of 3.6% (*n* = 2) and mean survival of 97.4 months (95% CI 89.3, 105.4 months).

Among the patients, 3.6% (*n* = 2) had renal failure requiring renal replacement therapy and 1.8% (*n* = 1) had respiratory failure requiring intubation. Median length of ICU stay was 1 day, and median length of hospital stay was 7 days.

The prognostic factors of mortality which remained statistically significant were etiology of MV disease (*p* = 0.01) and type of operation (*p* = 0.034). Primary mitral valve regurgitation showed the best outcome with mean survival of 104.5 months, compared with MS and previous failed MV surgery/redo which showed mean survival of 70 months and 63.4 months, respectively. In terms of types of operation, mitral valve replacement demonstrated poorer survival than mitral valve repair (HR 22.2, *p* = 0.015). None of the routine cases had severe LVEF impairment; hence, LVEF could not be assessed as a prognostic factor. In this cohort, S’ was not associated with survival (*p* = 0.746). This might be due to the smaller number of cases (after exclusion of urgent and emergency cases) to allow significant detection.

## 4. Discussion

Our study cohort showed that most of the mortalities of patients with PH who underwent MV and TV surgery happened within 30 days, after which the survival rate remained almost static, with a 1-year survival rate of 89.5% and a 5-year survival rate of 87.6%. Within this group, we identified five predictors of mortality (urgency of surgery, S’, type of operation, LVEF and etiology of MV disease), six predictors of post-operative respiratory failure (urgency of surgery, etiology of MV disease, S’, pre-op LVEF, RV size and TAPSE) and four predictors of post-operative renal failure (urgency of surgery, etiology of MV disease, S’ and pre-op LVEF).

PH is known to be associated with higher risks of morbidity and mortality in cardiac surgery compared with those without PH [12,13,14]. The optimal timing of surgery in this group of patients remains controversial. According to the European Society of Cardiology/European Association of Cardio-Thoracic Surgery (ESC/EACTS) guidelines, surgery should be offered to asymptomatic chronic severe primary mitral regurgitation (MR) patients with preserved LV function, if they have systolic pulmonary arterial pressure (sPAP) of more than 50 mmHg at rest (evidence class IIa) [10]. This is because current evidence suggests that symptomatic patients with PH have worse surgical outcomes [10]. The aim of this retrospective study was to investigate any potential risk factors which can further contribute to the negative surgical outcome in this group of patients. This will help us to risk stratify them pre-operatively.

S’ is used as a measure of RV function. Our cohort showed that patients with S’ <9.5 cm/s had higher rates of mortality, post-op respiratory failure and renal failure. This is consistent with the evidence that RV dysfunction predicts post-cardiac surgery morbidity and mortality [15,16]. Interestingly, there are many ECHO parameters which are used to assess RV function, such as TAPSE, RVFAC and MPI, which were not associated with post-op renal failure and mortality in our cohort. However, TAPSE of <1.7 cm was predictive of post-op respiratory failure.

Qualitative measurement of RV size can be easily incorporated into risk assessment for patients undergoing cardiac surgery. We found that dilated RV size is associated with respiratory failure in first-time mitral valve surgery and higher mortality rate in both first-time and redo mitral valve surgery. In patients with pulmonary hypertension, the RV goes through a series of changes to adapt to the high afterload in the pulmonary vasculature [17]. In the initial phase, the RV increases its contractility to maintain its output (coupling) by RV hypertrophy. This is followed by RV dilatation and ultimately RV uncoupling [17]. It is suggested that RV volume is an important measure of PH. Further investigations into ECHO parameters of RV function and RV size might help us to find the optimal timing for surgery in this group of patients.

LVEF is another ECHO parameter which we found to have an impact on mortality, post-op respiratory failure and renal failure. Patients with impaired LV function are more susceptible to low cardiac output state post-op and require inotropic support [18]; hence, they are more at risk of end organ failure such as respiratory and renal failure [19]. Both of these complications increase the length of hospital stay and indirectly increase risks of other morbidities such as hospital-acquired infections, and subsequently have an impact on mortality [20,21].

One other important factor to consider is the degree of TR. Long-standing pulmonary hypertension causes increase in RV size and tricuspid annular dilatation, which leads to severe TR. Severe TR usually suggests there is an irreversible component of pulmonary vascular disease [22]. Axtell et al. recently published a retrospective study demonstrating that surgery does not improve survival in patients with isolated severe TR [23]. The etiology of TR included in their series was a mixture of functional (59%), degenerative (6%), acquired (20%), pacemaker-induced (4%) and congenital (11%). Current ESC/EACTS guidelines recommend TV surgical intervention for patients with severe functional TR undergoing left-sided valve surgery [10].

Our results showed that MV repair has better survival outcomes than MV replacement, which is consistent with the literature. An analysis of a large prospective, multicenter, international registry of patients with degenerative MR showed that MV repair was associated with better operative mortality and long-term survival, as well as carrying less valve-related complications [24]. A meta-analysis comparing the short-term and long-term outcomes between MV repair and MV replacement in patients with ischemic MR showed that MV repair has better survival outcomes [25]. Therefore, it is widely agreed that MV repair should be performed whenever possible as it has better long-term outcomes [6,26,27].

Our cohort demonstrated that patients with secondary MR and MS have worse mortality and higher rates of post-op renal failure and respiratory failure compared with primary MR. Among secondary MR patients, 67% (*n* = 2) died within 30 days of surgery. Both of the patients were critically unwell and required emergency surgery, which could be the confounding factor. Further studies to compare only elective surgeries in both primary and secondary MR will better delineate the difference in surgical outcomes. Moreover, secondary MR is a disease of the left ventricle rather than the valve itself, and this could also be the reason for the difference in surgical outcome. The most common cause of MS is rheumatic fever. The changes in the rheumatic mitral valve include leaflet thickening and fibrosis, commissural fusion and calcific degeneration. These pathological changes are complex and difficult to repair; hence, most MS cases are treated with MV replacement [28]. Jiao et al. performed a retrospective study and reported no difference in survival between MV repair and MV replacement in patients with rheumatic MS after propensity score matching, but MV repair is associated with lower valve-related complications and better quality of life [28]. In our cohort, 88.9% (*n* = 8) of MS patients had mitral valve replacement. The higher mortality and morbidity rates in MS compared with primary MR could be due to the higher rate of MV replacement or perhaps due to the difference in the disease itself.

It is not a surprising finding that emergency cases showed the worst outcome, as they often present with more severe symptoms with hemodynamic compromise. We performed subgroup analysis of only routine cases and found that the outcomes of MV and TV surgery were still affected by the etiology of MV disease and whether patients had MV repair or replacement surgery.

The main limitations of this research are the relatively small sample size from a single center and the retrospective nature of the study. We attempted to eliminate as many confounding factors as possible by having strict inclusion and exclusion criteria. The calculation of pulmonary pressure was based on ECHO measurement of RV systolic pressure with the assumption that there is no RV outflow tract obstruction. Pre-operative right heart catheter study is not a routine practice for MV surgery.

## 5. Conclusions

Urgency of surgery, etiology of MV disease, pre-op LVEF, S’, RV size and TAPSE are prognostic factors of MV and TV surgical outcome in this small cohort of patients with PH. Based on this, we conclude that in routine cases of PH patients with primary MR, normal LVEF and normal RV function have acceptable morbidity and mortality rates. A larger prospective study is warranted to validate our results.

## Figures and Tables

**Figure 1 medicina-59-01103-f001:**
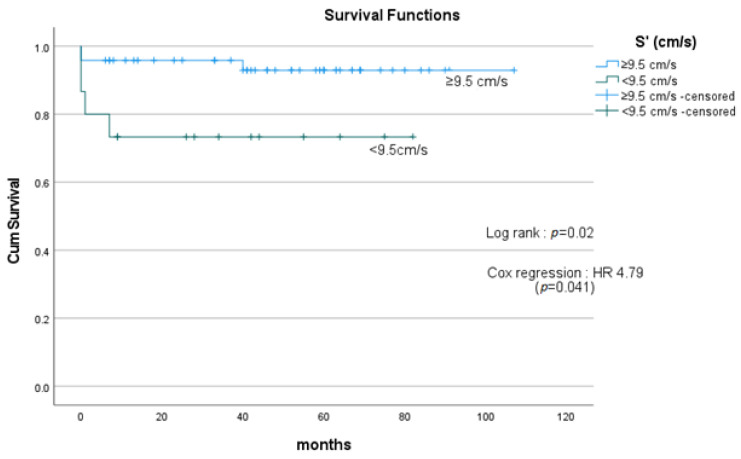
Survival curves comparing patients with S’ < 9.5 cm/s and ≥9.5 cm/s. Patients with S’ of <9.5 cm/s (*n* = 15) showed poorer survival compared with patients with S’ of ≥9.5 cm/s (*n* = 48), with mean survival of 57.4 months and 102.3 months, respectively; HR 4.79 (*p* = 0.041). HR = hazard ratio. S’= peak systolic tissue velocity at the tricuspid annulus.

**Figure 2 medicina-59-01103-f002:**
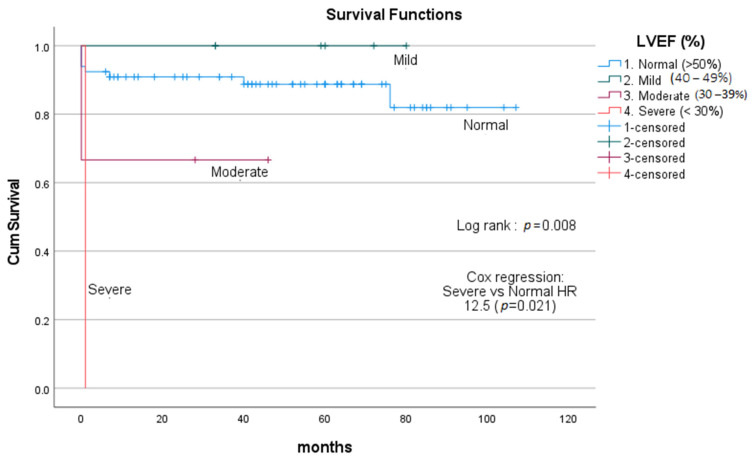
Survival curves comparing patients with different left ventricular ejection fraction (LVEF). Overall, there was a significant difference in survival among these 4 LVEF groups (*p* = 0.008). Cox regression only detected statistically significant HR between severe LVEF impairment and normal LVEF with HR of 12.5 (*p* = 0.021). HR = hazard ratio. LVEF = left ventricular ejection fraction.

**Figure 3 medicina-59-01103-f003:**
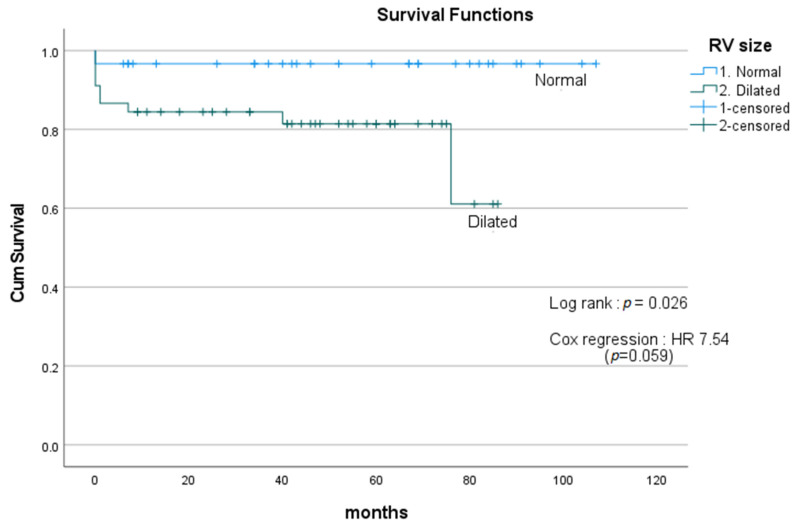
Survival curves comparing patients with normal and dilated RV. Dilated RV showed poorer survival compared with the normal RV group, with mean survival of 69.0 months and 103.2 months, respectively, using log-rank test (*p* = 0.026). However, further analysis with Cox regression did not detect any statistically significant difference with HR of 7.54 (*p* = 0.059). HR = hazard ratio. RV = right ventricle.

**Table 1 medicina-59-01103-t001:** Patient demographics and characteristics.

Demographics	Number (%)
1. Gender	
2. Female	41 (53.9)
3. Male	35 (46.1)
1. Pre-op AF	
2. No	37 (48.7)
3. Yes	39 (51.3)
1. Pre-existing CKD	
2. Yes	43 (56.6)
3. No	33 (43.4)
1. Pre-existing respiratory disease	
2. No	56 (73.7)
3. Yes	20 (26.3)
1. Etiology of MV disease	
2. Primary MR	55 (72.4)
3. Secondary MR	3 (3.9)
4. Mitral stenosis	9 (11.8)
5. Failed previous MV surgery	9 (11.8)
1. Etiology of tricuspid valve disease	
2. Primary TR	2 (2.6)
3. Secondary TR	74 (97.4)
1. Operation	
2. Mitral valve replacement + tricuspid valve repair	33 (43.4)
3. Mitral valve repair + tricuspid valve repair	41 (53.9)
4. Mitral valve replacement + tricuspid valve replacement	2 (2.6)
1. Degree of TR	
2. Mild	2 (2.6)
3. Moderate	25 (32.9)
4. Severe	49 (64.5)
1. RV dysfunction	
2. Normal	34 (44.7)
3. Mild	11 (14.5)
4. Moderate	18 (23.7)
5. Severe	13 (17.1)
1. RA volume index	
2. >28	51 (69.9)
3. ≤28	22 (30.1)
1. RVMPI	
2. <0.6	42 (62.7)
3. ≥0.6	25 (37.3)
1. RVFAC	
2. ≥0.35	16 (21.6)
3. <0.35	58 (78.4)
1. RV size (qualitative)	
2. Normal	30 (40)
3. Dilated	45 (60)
1. LVEF	
2. Normal	66 (86.8)
3. Mild	6 (7.9)
4. Moderate	3 (3.9)
5. Severe	1 (1.3)
1. S’	
2. <9.5 cm/s	15 (23.8)
3. ≥9.5 cm/s	48 (76.2)
1. TAPSE	
2. <1.7 cm	24 (34.3)
3. ≥1.7 cm	46 (65.7)

Abbreviations: AF (atrial fibrillation), CKD (chronic kidney disease), MV (mitral valve), MR (mitral regurgitation), TR (tricuspid regurgitation), RV (right ventricle), RA (right atrium), RVMPI (right ventricular myocardium performance index), RVFAC (right ventricle fractional area change), LVEF (left ventricle ejection fraction), S’ (peak systolic tissue velocity at the tricuspid annulus), TAPSE (tricuspid annular plane systolic excursion).

**Table 2 medicina-59-01103-t002:** Predictors of post-op renal failure and respiratory failure.

	Renal Failure, *n* (%)	Respiratory Failure, *n* (%)
1. LVEF	*p* = 0.01	*p* < 0.001
2. Hyperdynamic	0 (0)	0 (0)
3. Normal	5 (8.4)	2 (3.4)
4. Mild	0 (0)	1 (16.7)
5. Moderate	1 (33.3)	1 (33.3)
6. Severe	1 (100)	1 (100)
Urgency of surgery	*p* < 0.001	*p* = 0.002
1. Routine	2 (3.6)	1 (1.8)
2. Urgent	2 (13.6)	2 (13.3)
3. Emergency	3 (60.0)	2 (40.4)
1. S’	*p* < 0.001	*p* = 0.013
2. <9.5 cm/s	4 (26.7)	3 (20)
3. ≥9.5 cm/s	0 (0)	1(2.1)
1. Etiology of MV disease	*p* < 0.001	
2. Primary MR	2 (3.6)	1 (1.8)
3. Secondary MR	2 (66.7)	2 (66.7)
4. Mitral stenosis	3 (33.3%)	2 (22.2)
5. Failed previous MV surgery	0 (0)	0 (0)
1. TAPSE	*p* = 0.179	*p* = 0.025
2. <1.7 cm	4 (16.7)	4 (16.7)
3. ≥1.7 cm	3 (6.5)	1 (2.2)

Abbreviations: LVEF (left ventricle ejection fraction), S’ (peak systolic tissue velocity at the tricuspid annulus), MV (mitral valve), MR (mitral regurgitation), TAPSE (tricuspid annular plane systolic excursion).

**Table 3 medicina-59-01103-t003:** Summary of surgical outcome predictors.

Predictors	Predictor of Mortality	Predictor of Respiratory Failure	Predictor of Renal Failure
LVEF	Yes	Yes	Yes
S’	Yes	Yes	Yes
Urgency of surgery	Yes	Yes	Yes
Etiology of MV disease	Yes	Yes	Yes
Type of operation	Yes	No	No
Degree of TR	No	No	No
TAPSE	No	Yes	No
RV size			
Including redo MV surgery	No	No	No
Excluding redo MV surgery	No	Yes	No

Abbreviations: MV (mitral valve), TR (tricuspid regurgitation), LVEF (left ventricle ejection fraction), S’ (peak systolic tissue velocity at the tricuspid annulus), TAPSE (tricuspid annular plane systolic excursion).

## Data Availability

The data presented in this study are available on request from the corresponding author.

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
