# Peer review of "Predictors of Outcome in Patients with Pulmonary Hypertension Undergoing Mitral and Tricuspid Valve Surgery"

_medicina, 2023, doi:10.3390/medicina59061103_

Round 1

Reviewer 1 Report

An extensive study is presented with multiple clinical variables included. Seems that the research has been performed for the investigation of very complex objective such as:

to investigate the prognostic factors of surgical outcome in patients with PH undergoing mitral valve (MV) and tricuspid valve (TV) surgery, in order to risk stratify their management.

However the conclusions (shown below) do not have the sufficient support based on the findings described in the current submission and based on the statistical analysis performed in very small subgroups of patients.

 Conclusion: Etiology of MV disease, type of operation, S’, pre-op LVEF, RV size and TAPSE are prognostic indicators in patients with PH undergoing MV and TV surgery.

Therefore, it is highly recommended to revise the submission in order to improve the presentation of findings that would lead to clearly described conclusions supported by the results/findings. 

Only minor editing in English is recommended.

Author Response

Dear reviewer,

Thank you very much for taking the time to read through our manuscript. We really appreciate your suggestions and comments to improve our submission. 

Based on yours and another reviewer's comments, we have made the following changes to the conclusion:

Conclusion: Urgency of surgery, etiology of MV disease, type of operation (replacement or repair), S’, pre-op LVEF are prognostic indicators in this small cohort of patients with PH undergoing MV and TV surgery. Larger prospective study is warranted to validate our findings.

We have added a subgroup analysis of only routine cases. Please see the changes in the full manuscript below. 

Best wishes,

Ee Phui Kew

Reviewer 2 Report

The authors conducted a retrospective study, in which they investigated predictors of outcome in patients with pulmonary hypertension undergoing mitral and tricuspid valve surgery. In the study, they showed that etiology of mitral valve disease, type of operation, S’, pre-ope LVEF., RV size and TAPSE are prognostic indicators in patients in patients with PH undergoing MV and TV surgery. The topic is important and the manuscript is well written. The following are my comments to improve the manuscript.

Major comments

1. Inclusion and exclusion criteria of the study should be written in detail. Especially, whether emergency surgery and infectious endocarditis were included or excluded in this study is important information. This is because these two conditions could affect the outcomes. In addition to this, presenting data regarding severity of preoperative PH is desirable for external validity.

2. The definition of respiratory failure and post-operative renal failure should be made clear.

3. What were the causes of mortality? If there are data available, I recommend to summarize these data.

Minor comments

1. Page5, Table 1

Two cases (2.6%) were diagnosed as mild TR.

Were these two cases underwent TVP/TVR?

2. As for follow-up, how many patients were dropped out?

Author Response

Dear reviewer,

Thank you very much for taking the time to read through our manuscript. We really appreciate your suggestions and comments to improve our submission. Below is our respond to your comments:

  1. Inclusion and exclusion criteria of the study should be written in detail. Especially, whether emergency surgery and infectious endocarditis were included or excluded in this study is important information. This is because these two conditions could affect the outcomes. In addition to this, presenting data regarding severity of preoperative PH is desirable for external validity.

Inclusion criteria for this study were:

  • Adult older than 18 years old.
  • Patients who underwent both mitral and tricuspid valve surgery in our unit between Jan 2011 and Dec 2019, including elective, urgent and emergency cases.
  • Proven pulmonary hypertension based on pre-operative echocardiogram.

Exclusion criteria:

  • Any concomitant cardiac surgeries such as coronary artery bypass surgery, aortic valve surgery, ascending aorta surgery.
  • Free flow TR on echocardiogram
  • Infective endocarditis as different pathology

In this cohort, there are 56 routine cases, 15 urgent cases and 5 emergency cases. Subgroup analysis performed on routine cases (n=56) showed all-cause mortality of 8.9% (n=5), in hospital mortality of 3.6% (n=2), with mean survival of 97.4 months (95% CI 89.3, 105.4 months).

3.6% (n=2) had renal failure requiring renal replacement therapy and 1.8% (n=1) had respiratory failure requiring intubation. Median length of ICU stay is 1 day and median length of hospital stay is 7 days.

The prognostic factors of mortality which remained statistically significant are etiology of MV disease (p=0.01) and type of operations (p=0.034). Primary mitral valve regurgitation showed the best outcome with mean survival of 104.5 months, compared to MS and previous failed MV surgery/redo which showed mean survival of 70 months and 63.4 months respectively. In terms of types of operation, mitral valve replacement demonstrated poorer survival than mitral valve repair (HR 22.2, p=0.015).    None of the elective patients had severe LVEF impairment, hence LVEF could not be assessed as a prognostic factor. In this cohort, S’ was not associated with survival (p=0.746). This might be due to the smaller number of cases (after exclusion of urgent and emergency cases) to allow significant detection.

  1. The definition of respiratory failure and post-operative renal failure should be made clear.

In this study, the definition of respiratory failure is the need for re-intubation during the same admission, and the definition for renal failure is the need for renal replacement therapy (either continuous veno-venous haemofiltration or haemodialysis) during the same admission.

  1. What were the causes of mortality? If there are data available, I recommend to summarize these data.

Unfortunately we do not have the exact cause of mortality.

Minor comments

  1. Page5, Table 1

Two cases (2.6%) were diagnosed as mild TR.

→Were these two cases underwent TVP/TVR?

Those two cases did undergo tricuspid annuloplasty. In one of the cases, the operating surgeon stated that there was gross annular dilation, hence decision was made to perform annuloplasty. In the second case, there was no clear documentation regarding the intra-operative findings but TV annuloplasty was performed.

Please see the revised manuscript below. Thank you for your time.

Best wishes,

Ee Phui Kew

  1. As for follow-up, how many patients were dropped out?

There was no drop out in this study. All the patients were followed up until mortality/or the end of the study period.

Round 2

Reviewer 2 Report

Thank you for the revision. The authors answered almost all of my comments and the manuscript has improved. The only one comment left unanswered is the following.

Presenting data regarding severity of grading of preoperative pulmonary hypertension is desirable for external validity. For instance, data on patients stratified by mean pulmonary artery pressure estimated with echocardiography. (e.g. Mild=20-40mmHg, Moderate=41-55mmHg. Severe=>55mmHg).

Author Response

Dear reviewer,

Thank you very much for taking the time to review our submission. Below is our response your previous comment regarding pulmonary hypertension:

Pulmonary hypertension was assessed with ECHO measurement of RV systolic pressure with the assumption that there is no RV outflow tract obstruction. Pre-operative right heart catheter study is not a routine practice for MV surgery. 9.2% (n=7) had mild PH (RVSP 35 – 45mmHg), 39.5% (n=30) had moderate PH (RVSP 46-60mmHg), and 51.3% (n=39) had severe PH (RVSP >60mmHg). The degree of PH in this cohort did not affect their survival (p=0.374).

Please let us know if you have any further questions. We have made the changes in our manuscript and it is attached below.

Best wishes,

Ee Phui Kew
